# Insights into Pathogenesis, Nutritional and Drug Approach in Sarcopenia: A Systematic Review

**DOI:** 10.3390/biomedicines11010136

**Published:** 2023-01-05

**Authors:** Rodrigo Haber Mellen, Otávio Simões Girotto, Eduarda Boni Marques, Lucas Fornari Laurindo, Paulo Cesar Grippa, Claudemir Gregório Mendes, Lorena Natalino Haber Garcia, Marcelo Dib Bechara, Sandra Maria Barbalho, Renata Vargas Sinatora, Jesselina Francisco dos Santos Haber, Uri Adrian P. Flato, Patricia Cincotto dos Santos Bueno, Claudia Rucco Penteado Detregiachi, Karina Quesada

**Affiliations:** 1Department of Biochemistry and Pharmacology, School of Medicine, University of Marília (UNIMAR), São Paulo 17525-902, Brazil; 2Postgraduate Program in Structural and Functional Interactions in Rehabilitation—University of Marília (UNIMAR), São Paulo 17525-902, Brazil; 3School of Food and Technology of Marilia (FATEC), São Paulo 17590-000, Brazil; 4Department of Animal Sciences, School of Veterinary Medicine, University of Marília (UNIMAR), São Paulo 17525-902, Brazil

**Keywords:** sarcopenia, inflammation, amino acids, drugs

## Abstract

Sarcopenia is a multifactorial condition related to the loss of muscle mass and strength due to aging, eating habits, physical inactivity, or even caused by another disease. Affected individuals have a higher risk of falls and may be associated with heart disease, respiratory diseases, cognitive impairment, and consequently an increased risk of hospitalization, in addition to causing an economic impact due to the high cost of care during the stay in hospitals. The standardization of appropriate treatment for patients with sarcopenia that could help reduce pathology-related morbidity is necessary. For these reasons, this study aimed to perform a systematic review of the role of nutrition and drugs that could ameliorate the health and quality of life of sarcopenic patients and PRISMA guidelines were followed. Lifestyle interventions have shown a profound impact on sarcopenia treatment but using supplements and different drugs can also impact skeletal muscle maintenance. Creatine, leucine, branched-chain amino acids, omega 3, and vitamin D can show benefits. Although with controversial results, medications such as Metformin, GLP-1, losartan, statin, growth hormone, and dipeptidyl peptidase 4 inhibitors have also been considered and can alter the sarcopenic’s metabolic parameters, protect against cardiovascular diseases and outcomes, while protecting muscles.

## 1. Introduction

Sarcopenia, from the Greek *sarx* for “meat” and *penia* for “loss” is a multifactorial condition related to the loss of muscle mass due to aging, eating habits, physical inactivity, or even caused by another disease [1,2]. Currently, it has been discussed with a focus on muscle function, such as muscle strength, instead of only based on muscle mass [3,4].

The pathophysiology of sarcopenia is related to the imbalance between the production of muscle cells and their degradation. With aging, the amount of muscle cells is gradually depleted, and anabolic stimuli create resistance with a consequent decrease in mitochondrial function, changes in gene expression, reduced insulin sensitivity, and impaired neuromuscular signaling [5,6,7,8]. On the other hand, with age, there is a relationship associated with reduced secretion of the hormones that are related to the stimulation of skeletal muscle mass and function, such as growth hormone (GH), insulin-like growth factor 1 (IGFI), testosterone, and estradiol. Furthermore, myogenesis can be affected by altered IGF-I signaling along with decreased insulin sensitivity. Other hormones decrease with advancing age, such as adrenal-derived dehydroepiandrosterone, thyroid hormones, and vitamin D, which may also be related to the pathophysiological process of sarcopenia. Adipokines released by adipose tissue produce important modifications during aging and can affect muscle metabolism and physiology. Therewithal, catabolic hormones such as cortisol and angiotensin II can accelerate age-induced muscle atrophy as they are involved in muscle wasting, and their levels increase with age [9,10,11,12].

Affected individuals have a higher risk of falls, the ability to perform simple activities is impaired, and may be associated with heart disease, respiratory diseases, cognitive impairment, and consequently an increased risk of hospitalization, in addition to causing an economic impact due to the high cost of care during the stay in hospitals [13,14]. However, treatment for sarcopenic patients can be non-drug and drug-based. In the latter category, there is no consensus on a single drug to be used that is currently approved for the therapeutic approach of sarcopenia [15]. On the other hand, in the nutritional aspect, the hypertrophic response can be increased by amino acid supplementation or protein-rich diets. Moreover, the use of supplementation such as creatine can also contribute to this response [1,16]. The practice of physical exercise, aerobic or muscular resistance, and a balanced diet are indispensable for improving the disease, not only for muscular hypertrophy but also for the metabolic and energetic maintenance of the skeletal system. In some patients, the administration of hormones is also used to increase muscle mass [15,17]. Figure 1 highlights the major risk factors related to sarcopenia development and its consequences.

Increasing awareness and understanding of the disease are of paramount importance for standardizing appropriate treatment and diagnostic options that will lead to better care of patients with sarcopenia, aiming at decreasing pathology-related morbidity. Therefore, the objective of this study is to perform a systematic review to investigate the nutritional and drug aspects that could improve the health and quality of life of sarcopenic patients.

## 2. Materials and Methods

### 2.1. Focal Question

The focal question for this review was: what are the nutritional and drug aspects that could improve sarcopenic patients’ health and quality of life?

### 2.2. Databases

For this review, we consulted MEDLINE-PubMed, Cochrane, and EMBASE databases. The search for the studies followed Preferred Reporting Items for a Systematic Review and Meta-Analysis (PRISMA) guidelines [18]. Figure 2 shows the selection of the studies.

The combination of mesh terms searched in the databases were sarcopenia or loss of muscle mass and aging or inflammation or oxidative stress or pathogenesis or nutrition or supplementation or drug or medication. There was no restriction for the search period. Based on the studies resulting from the mesh terms, the discussion was built.

### 2.3. Eligible Criteria

Eligibility criteria for the literature search for human studies were based on the PICO (Population, Intervention, Comparison, and Outcomes) format. Studies in animal models were also included.

### 2.4. Study Selection

Only studies published in English were included. Inclusion criteria were investigative studies, clinical trials, and randomized clinical trials (RCT).

The exclusion criteria were articles not published in English, not full-text articles, letters to the editor, case reports, and poster presentations.

Two independent authors (KQ, and RHM) performed the search in the databases and independently retrieved the studies that followed the mesh terms. Full-text articles were studied to support decision-making. The other two reviewers evaluated disagreements between the two reviewers (SMB and MDB).

The outcomes evaluated (mean difference) were improvement in physical and functional performance, improvement in walking time, and increased lean mass index.

### 2.5. Extraction of Data

Data were retrieved from eligible articles, including the author, date, study design, information associated with sarcopenia, pathophysiology, nutrition, and drug approach. 

### 2.6. Evaluation of the Risk of Bias in the Included Studies

The risk of biases in the included studies was carefully considered and analyzed according to each study’s selection, detection, and reporting of bias. Risks of quality and bias in the inclusion of subjects/patients, interventions, evaluation of outcomes, missing information, and data were also investigated. This evaluation of the risk of bias and quality followed the directives of the Cochrane Handbook for Systematic Reviews of Interventions [19].

## 3. Results

The search performed in the databases resulted in the selection of studies related to the use of creatine, leucine, branched-chain amino acids, omega 3, and vitamin D. These studies showed that the supplementation with these substances can show benefits in the therapeutic approach of sarcopenic patients. Clinical trials were found only for the use of leucine [20,21], BCAA [22,23,24], omega-3 [25,26], and calcium [27,28,29,30]. These trials are summarized in Table 1. 

Although with controversial results, medications such as SGLT inhibitors [31] growth hormone [32], glucagon-like peptide-1 [33], metformin [34,35], 3-Hydroxy-3 methylglutaryl coenzyme A inhibitor [36,37,38], and dipeptidyl peptidase 4 inhibitors [39,40,41] can also produce beneficial effects altering the sarcopenic’s metabolic parameters, protecting against cardiovascular diseases and outcomes while protecting muscles. These trials are shown in Table 2.

The risk of bias is shown in Table 3 and Table 4.

**Table 1 biomedicines-11-00136-t001:** Clinical studies showing the use of dietary supplements in the approach of sarcopenia.

Reference	Population	Intervention/Comparison	Outcomes
Leucine			
[20]	Randomized, single-blind, placebo-controlled study (Spain) with 41 post-hospitalized older adults (82.1 ± 5.89 y) randomly divided into leucine + RTI (82.9 ± 5.67 y, 19♂ and 22♀) and placebo + RTI (81.2 ± 6.14 y, 10♂ and 10♀) groups	20 g of whey protein isolated +3 g of leucine/2 non-consecutive days per week	↑Physical performance, ↓frailty, and ↑myostatin (↑appendicular skeletal muscle mass) but not significantly between intervention and placebo
[21]	Randomized, double-blind, placebo-controlled study (Spain) with 50 elder (78.9 ± 7.9 y) living in nursing homes and able to walk 6 m were randomly divided into leucine (with MHS of 16.3 ± 8.5 kg) and placebo (with MHS of 19.2 ± 8.6 kg) groups	6 g/day of leucine/13 weeks	↑Functional performance, ↑walking time, ↑lean mass index, and ↑maximum static expiratory force
BCAA			
[22]	Randomized, double-blind, placebo-controlled study (India) with 60 sarcopenic viral and alcohol-related cirrhosis patients (41.6 ± 9.9 y) randomly divided into BCAA (42.26 ± 10.07 y, 19♂ and 11♀) and placebo (40.83 ± 9.80 y, 21♂ and 9♀) groups	12 g/day of BCAA/6 months	↑Muscle mass, ↑MHS, ↑6-min walk distance, and ↑6-m gait speed but not significantly between intervention and placebo
[23]	Retrospective observational study (Japan) with 29 stroke patients divided into LEBDs (77–92 y, 6♂ and 9♀) and SBDs (77.8–86.3 y, 5♂ and 9♀) groups	LEBDs (2.07 g of BCAA, 1.44 g of leucine, 0.36 g of valine and 0.27 g of isoleucine) or SBDs (1.58 g of BCAA, 0.72 g of leucine, 0.48 g of valine and 0.38 g of isoleucine)/Patients received the intervention twice a day on the fifth and seventh days of hospitalization	Improvements in transthyretin and CRP were observed, but not significantly between the groups
[24]	Randomized, single-blind study (Japan) with 66 stroke patients randomly divided into breakfast (65.5 ± 13.1 y, 14♂ and 9♀) and post-exercise (67.5 ± 5 y, 14♂ and 9♀) groups	3.5 g of amino acids and 6.5 g of protein + 40 IU of VD/day/2 months	↑Leg press strength, ↑physical performance, ↓body fat mass, and improvement in Berg balance scale but without significant timing influence
Omega 3			
[25]	Randomized, double-blind, placebo-controlled study (Belgium) with 23 older adults (65–83 y, 15♂ and 8♀) randomly divided into PUFA (intervention) or PLAC (placebo) groups	1100 mg ω-3 soft gels (1020 mg ω-3 + 410 mg DHA + 540 mg EPA + 4 mg vitamin E) 3× daily/14 weeks	↑Knee-extensor strength and synergism with resistance training in improving muscle inflammatory and catabolic markers (FOXO1 and LC3b)
[26]	Population-based cross-sectional study (Iran) with 300 elderly adults (150♂ and 150♀, ≥65 y) were studied due to their eating habits	Anti-inflammatory (omega 3 + other nutrients) or other diets	Patients with anti-inflammatory diet presented lower odds of sarcopenia
Calcium			
[27]	Randomized, double-blind, placebo-controlled study (Lebanon) with 248 overweight adults (55%♀, 71 ± 4.6 y, 30.2 ± 4.5 Kg/m^2^ of BMI) with baseline VD of 10–30 ng/mL randomly divided into VD and placebo groups	3750 IU/day of VD/12 months	There were no improvements in the indices of sarcopenia or adiposity between the groups
[28]	Randomized, double-blind, placebo-controlled study (Belgium) with 15 adult patients with thermal burns dating from 2 to 5 years randomly divided into calcium + VD (22–58 y, 7♂ and 1♀) and placebo (29–64 y, 4♂ and 3♀) groups	Quarterly IM injection of 200,000 IU of VD + daily oral calcium/12 months	↑Quadriceps strength when tested at high velocity significantly but without significant improvements in bone health
[29]	Randomized, double-blind, placebo-controlled study (United States). [Phase 1] NE_PLA_ (14♂ and 11♀, 72 ± 1 y) and NE_HMB_ (13♂ and 12♀, 73 ± 1 y) older adults groups; [Phase 2] RE_PLA_ (11♂ and 13♀, 73 ± 1 y) and RE_HMB_ (11♂ and 13♀, 73 ± 1 y) older adults groups.	[Phase 1] 3 g of CaHMB twice daily and [Phase 2] 3 g of CaHMB twice daily + RTI/24 weeks [Phase 1] and 24 weeks [Phase 2]	CaHMB significantly improved muscle strength and MQ independently of RTI
[30]	Prospective cohort study (Australia) with 740 non-institutionalized older adults (50%♀ and with mean age pf 62 ± 7 y) randomly sampled	Patients were analyzed for their nutrient intake at baseline and follow-up (2.6 ± 0.4 y later)	Significant positive associations were found between calcium intake and aLM

↑, increase; ↓, decrease; ω-3, omega 3; aLM, appendicular lean mass; BCAA, branched-chain amino acids; BMI, body mass index; CaHMB, calcium β-hydroxy-β-methylbutyrate; CRP, c reactive protein; DHA, docosahexaenoic acid; EPA, eicosapentaenoic acid; FOXO1, forkhead box protein O1; LC3b, microtubule-associated protein 1A/1B-light chain 3; LEBDs, leucine enriched BCAA dietary supplement; MHS, muscular handgrip strength; MQ, muscle quality; NE_HMB_, ad libitum diet plus CaHMB; NE_PLA_, ad libitum diet plus placebo; RE_HMB_, ad libitum diet plus CaHMB and resistance exercise; RE_PLA_, ad libitum diet plus placebo and resistance exercise; RTI, resistance training intervention; SBDs, standard BCAA dietary supplement; VD, vitamin D; y, years.

**Table 2 biomedicines-11-00136-t002:** Clinical trials that investigated the use of medications in the therapeutic approach of sarcopenia.

Reference	Population	Intervention/Comparison	Outcomes
SGLT2 inhibitors			
[31]	Prospective cohort study (Japan) with 43 moderately obese Japanese patients (53.5 ± 8.04 y, 27♂ and 10♀) with T2DM treated with luseogliflozin	Personalized doses of luseogliflozin */52 weeks *	↓Body fat and ↑skeletal muscle mass significantly
GH analogs			
[32]	Secondary analyses of two previously completed randomized clinical trials (United States) with 341 individuals living with HIV and with abdominal obesity divided into tesamorelin (47.8 ± 7.3 y, 89.1%♂) and placebo (48 ± 7.6 y, 83.8%♂) groups	Personalized doses of tesamorelin/26 weeks *	Tesamorelin treatment significantly increased skeletal muscle mass, area, and density in those patients with significant decreases in visceral adipose tissue
GLP-1A			
[33]	Prospective cohort study (Italy) with 6♂ (68.50 ± 4.23 y, 135.83 ± 31.38 mmol/l of FBG) and 3♀ (67.66 ± 3.78 y, 173.00 ± 49.24 mmol/l of FBG)	Liraglutide 3 mg/day/24 weeks	↓Body fat mass and ↑muscle tropism protecting against sarcopenia
Metformin			
[34]	Cross-sectional observational study (China) with 1427 (535♂ and 892♀) individuals with (504♂ and 775♀) and without (31♂ and 117♀) sarcopenia	Personalized doses of metformin *	Metformin was effective in protecting against muscle mass loss among T2DM individuals
[35]	Randomized, double-blind, placebo-controlled clinical study (Indonesia) with 91 non-diabetic elderly individuals randomly divided into metformin (67.77 ± 5.14 y, 19♂ and 24♀) and placebo (70.04 ± 5.34 y, 15♂ and 33♀) groups	500 mg thrice daily/16 weeks	↑Usual gait speed significantly but did not improve handgrip strength or myostatin levels
3-Hydroxy-3 methylglutaryl coenzyme a inhibitors (statins)			
[36]	Prospective cohort study (Finland) with 216 abdominal aortic aneurysms patients that underwent EVAR (77.7 ± 7.4 y, 188♂ and 28♀) divided into statin users (77.4 ± 7.5 y, 113♂ and 16♀) or nonusers (78.2 ± 7.3 y, 75♂ and 12♀)	10 to 80 mg/day of atorvastatin, rosuvastatin, simvastatin orfluvastatin/At least a 4-month period of statin pre-treatment before EVAR	Statin treatment decreased long-term mortality among patients that underwent EVAR without pre-disposing to increased risk for sarcopenia
[37]	Population-based nationwide retrospective cohort study (Taiwan) with 67,001 clinically confirmed cases of CKD (2407 with sarcopenia) divided into statins users (547 with sarcopenia) and nonusers (1860 with sarcopenia)	Personalized doses of pravastatin, fluvastatin, atorvastatin, lovastatin, simvastatin or rosuvastatin *	Patients with CKD could receive statins treatment to reduce individual’s risk of developing newly diagnosed sarcopenia
[38]	Prospective cohort study (United Kingdom) with 639 older adults (321♂ and 318♀ with 64.1 ± 2.5 y and 65.9 ± 2.7 y, respectively) that were undergoing statins, thiazides or ACE inhibitors treatments	Personalized doses of statins, thiazides or ACE inhibitors/Mean follow-up time was about 4.4 y *	Any treatment was associated significantly with protection against handgrip strength decline
DPP-4 inhibitors			
[39]	Retrospective cohort study (Turkey) with 90 T2DM geriatric patients (72.57 ± 7.089 y, 60%♀) divided into DPP4 users (*n* = 48, 72.88 ± 7.13 y) and nonusers (*n* = 42, 72.21 ± 7.10 y)	Personalized doses of DPP-4 inhibitors/6 months *	DPP-4 inhibitors therapy was effective in improving muscle strength among geriatric T2DM patients
[40]	Retrospective observational study (Japan) with 105 T2DM patients (62 ± 12 y, 39%♀) divided into DPP-4 inhibitors users (64 ± 13 y, 49%♂) and nonusers (60 ± 12y, 68%♂)	Personalized doses of statins, thiazides or DPP-4 inhibitors *	Among DPP-4 inhibitors users, the skeletal muscle index was significantly higher in comparison with nonusers
[41]	Cross-sectional cohort study (Italy) with 80 elderly diabetic patients (76.2 ± 5.4 y, 38♂ and 42♀) treated with DPP-4 (74.9 ± 4.8y, 17♂ and 20♀) or sulfonylureas (77.1 ± 5.3 y, 21♂ and 22♀) for at least 24 months	Personalized doses of statins, thiazides or DPP-4 inhibitors *	DPP-4 users had significant improvements in sarcopenia parameters such as fat-free mass decrease, skeletal muscle mass increase, and increases in muscle strength and gait speed

* Personalized doses of the medications (not mentioned in the study); ↑, increase; ↓, decrease; ACE, angiotensin converting enzyme; CKD, chronic kidney disease; DPP-4, dipeptidyl peptidase 4; EVAR, endovascular repair; FBG, fasting blood glucose; GH, growth hormone; GLP-1A, glucagon-like peptide-1 receptor agonists; HIV, human immunodeficiency virus; SGLT2, sodium-glucose cotransporter 2; T2DM, type 2 diabetes mellitus; y, years.

**Table 3 biomedicines-11-00136-t003:** Risk of bias of the clinical studies showing the use of dietary supplements in the approach of sarcopenia.

Reference	Question Focus	Appropriate Randomization	Allocation Blinding	Double-Blind	Losses (˂20%)	Porgnostics or Demographic Carachteristics	Outcomes	Intestion to Treat Analyses	Sample Calculation	Adequate Follow-Up
Leucine										
[20]	Yes	No	Yes	No	Yes	Yes	Yes	No	Yes	Yes
[21]	Yes	NR	Yes	Yes	Yes	No	Yes	No	Yes	Yes
BCAA										
[22]	Yes	Yes	Yes	Yes	Yes	Yes	Yes	Yes	Yes	Yes
[23]	Yes	No	No	No	No	Yes	Yes	No	NR	No
[24]	Yes	Yes	Yes	No	Yes	No	Yes	Yes	Yes	Yes
Omega 3										
[25]	Yes	No	Yes	Yes	NR	No	Yes	NR	NR	Yes
[26]	Yes	No	No	No	Yes	Yes	Yes	No	Yes	No
Calcium										
[27]	Yes	No	Yes	Yes	Yes	Yes	Yes	No	Yes	Yes
[28]	Yes	Yes	Yes	No	Yes	Yes	Yes	No	No	Yes
[29]	Yes	Yes	Yes	Yes	Yes	Yes	Yes	No	NR	Yes
[30]	Yes	No	No	No	No	Yes	Yes	No	NR	Yes

NR, not reported.

**Table 4 biomedicines-11-00136-t004:** Risk of bias of the clinical trials that investigated the use of medications in the therapeutic approach of sarcopenia.

Reference	Question Focus	Appropriate Randomization	Allocation Blinding	Double-Blind	Losses (˂20%)	Porgnostics or Demographic Carachteristics	Outcomes	Intestion to Treat Analyses	Sample calculation	Adequate Follow-Up
SGLT2 inhibitors										
[31]	Yes	No	No	No	Yes	Yes	Yes	No	NR	Yes
GH analogs										
[32]	Yes	NR	Yes	No	NR	Yes	Yes	NR	NR	Yes
GLP-1A										
[33]	Yes	No	No	No	Yes	Yes	Yes	Yes	NR	Yes
Metformin										
[34]	Yes	No	No	No	Yes	Yes	Yes	Yes	NR	Yes
[35]	Yes	No	Yes	Yes	No	Yes	Yes	No	Yes	Yes
3-Hydroxy-3 methylglutaryl coenzyme a inhibitors (statins)										
[36]	Yes	No	No	No	NR	Yes	Yes	NR	No	Yes
[37]	Yes	No	No	No	Yes	Yes	Yes	No	NR	Yes
[38]	Yes	No	No	No	Yes	Yes	Yes	NR	NR	Yes
DPP-4 inhibitors										
[39]	Yes	No	No	No	Yes	Yes	Yes	Yes	Yes	Yes
[40]	Yes	No	No	No	Yes	Yes	Yes	Yes	Yes	Yes
[41]	Yes	No	No	No	Yes	Yes	Yes	Yes	Yes	Yes

NR, not reported.

## 4. Discussion

### 4.1. Sarcopenia: Pathophysiological Aspects

The age-related muscle mass loss and function in sarcopenia is closely linked to the expressive reduction of motoneurons innervating muscle fibers. Besides that, there are post-translational alterations of muscle proteins and a lack of coordination between the expression of contractile proteins, mitochondria, and sarcoplasmic reticulum in the intracellular environment [42,43]. Skeletal muscle homeostasis is impacted by an imbalance between anabolism and protein catabolism resulting in generalized muscle loss, muscle strength, and physical performance [4]. Changes in the sarcopenic muscle decrease the size and number of myofibrils, particularly affecting type II fibers. This occurs due to the transition from type II to type I fibers, fat infiltration between and within the muscle, and a reduction in satellite cells that are fundamental in muscle regeneration [13,44].

There is a relationship between adipose tissue and muscle atrophy associated with a decrease in the expression of human contractile proteins in myotubes when co-cultured with adipocytes extracted from individuals with high BMI. When the subject is obese and also possesses metabolic syndrome (MetS), the adipocytes can suffer modifications in the secretory pattern, expressing pro-inflammatory mediators due to infiltration of activated pro-inflammatory macrophages (M1) and other immune cells [45,46,47]. The result is an increase in the production of leptin (due to the resistance to the action of this hormone), resistin, tumor necrosis factor-alpha (TNF-α), interleukin (IL)-6, IL-18, plasminogen activator inhibitor (PAI-1), and reduction in anti-inflammatory mediators such as IL-10. Modified adipose tissue cells are characterized by a high production of circulating pro-inflammatory molecules, which have degrading effects in tissues such as the hypothalamus, liver, pancreas, and muscles [45,48,49,50,51]. 

Reactive oxygen (ROS) and nitrogen (RNS) species cause oxidative changes and damage to DNA, proteins, and lipids, where their function and structure are altered. With this, the functioning of antioxidant systems is impaired as these oxidized molecules accumulate and spread into the tissue, causing aging. Therefore, the aging process can be recognized as a factor that drives the accumulation of ROS, mainly due to the decrease in cellular antioxidant activities that lead to the accumulation of free radicals. Considering that muscle loss usually occurs due to a combination of muscle atrophy and muscle cell death, the death of these cells may be related to the uncontrolled production of free radicals [52,53,54]. 

Skeletal muscle easily adapts to different stimuli, even overproduction of ROS and RNS. Because skeletal muscle has a high metabolic rate, a greater predisposition to accumulate oxidized molecules, and a decreased response from antioxidant systems, older adults are at greater risk of delayed muscle regeneration, which can lead to sarcopenia. This is due to an alteration in the mitochondrial DNA (mtDNA) mediated by ROS that causes a dysfunction in the respiratory chain, further increasing the production of these species and establishing a progressive respiratory dysfunction in the mitochondria, which leads to induced cell death. During a state of stress, mitochondria can produce peptides, known as myokines, that affect other cells and contribute to the rate of aging [54,55,56,57]. 

Sarcopenia decreases muscle mass/strength and reduces muscle regeneration capacity caused by inflammatory processes, mainly due to a lack of physical exercise. This scenario induces a change in the neurohormonal response, leading to an imbalance in the production and degradation of proteins. The inflammatory process is mediated by decreased satellite cells responsible for muscle growth, maintenance, and repair. Therefore, this pro-inflammatory and anti-inflammatory imbalance results in chronic muscle inflammation that can cause damage, such as muscle deterioration [58].

Excessive generation of ROS and RNS can cause metabolic and contractility changes. In muscles, the greatest energy demand occurs during contraction. Producing this energy in mitochondria is also related to the release of superoxide anion radicals, which is also related to the synthesis of other free radicals. However, these free radicals produced in contraction trigger an adaptive response. Activating transcription factors will modulate the gene expression of antioxidant enzymes, such as superoxide dismutase (SOD), helping eliminate the superoxide anion. Several pathways generate excessive superoxide anion, including the mitochondrial electron transport chain, NAD(P)H oxidase, and endothelial uncoupled nitric oxide synthase, leading to cell and tissue damage in various organs. SOD converts superoxide to hydrogen peroxide and molecular oxygen to protect organs from oxidative stress. Then hydrogen peroxide is converted to water by catalase or glutathione peroxidase. Thus, SOD plays a critical role in the antioxidant defense system and serves as a defender against various diabetic complications, including sarcopenia [59,60,61].

Furthermore, oxidative stress through proteolytic pathways, pro-inflammatory cytokines, and the release of pro-apoptotic leads to protein breakdown and impairment in protein synthesis/apoptosis. Among the most important markers are glycoprotein dikkopf-3 (Dkk-3), associations of plasma markers with 8-isoprostane, and micro-RNAs (miR-21, miR-206, and miR-133) [44]. TNF-α promotes protein degradation and may be associated with obesity-related skeletal muscle atrophy and age-related sarcopenia. It induces atrophic factors like MAFbx/atrogin-1 and MuRF1 in myotubes and causes muscle wasting by inducing the ubiquitin-proteasome system [45].

Altering mitochondrial function and biogenesis by increasing ROS production generates more inflammation, promoting increased activity of NLRP3 (pyrin domain of the Nod 3-like receptor family) that participates in sarcopenia [54,62,63,64]. TNF-α activates local vascular endothelial cells, which causes the release of NO (nitric oxide), resulting in increased vascular permeability, allowing the passage of pro-inflammatory cells, and triggering excessive inflammation. The increase in TNF-α levels is related to lower muscle mass, contributing to sarcopenia. This is also correlated with the activation of apoptosis in muscle cells. In addition, high interleukin-6 (IL-6) plasma levels have consequences related to fatigue and disability associated with muscle destruction. However, it appears that these two markers favor the appearance of sarcopenia and increase the production of acute-phase reactive proteins by the liver, such as C-reactive protein (CRP) or α1-antichimotrypsin (ACT). On the other hand, acute muscle regeneration is modulated by macrophages, and a greater degree of neutrophilia affects the transition of the appropriate temporal macrophage and is associated with a decrease or even inhibition of regenerative muscle capacity [58,65,66]. Moreover, it is worth mentioning that there is a close relationship between muscle fragility and sarcopenia, as observed in patients with multimorbidity, such as obesity, diabetes mellitus, atherogenic dyslipidemia, and systemic arterial hypertension, in which aging and chronic conditions can lead to extensive oxidative stress, inflammation, and apoptosis [67,68,69].

Figure 3 shows the potential crosstalk between chronic inflammation and hormone function dysregulations in the development of sarcopenia.

#### Diseases Related to Sarcopenia

Patients with peripheral arterial disease (PAD) are more likely to develop sarcopenia through an altered inflammatory process, probably mediated by the deregulation of multiple cytokine-related factors, such as the elevation of IL-6, an IL-1 receptor antagonist, fibrinogen, and CRP. Studies show a relationship between increases in these markers and decreases in walking ability and overall exercise performance [70,71]. In PAD, ischemic lesions generate extensive oxidative stress. Murine models of PAD exhibited reduced levels of mRNA of antioxidant enzymes, increased production of ROS, and elevation of oxidative stress. Mitochondrial respiration is impaired, closely linked with the diminished activities of complexes I, III, and IV of the electron transport chain in ischemic muscles. Likewise, human studies have shown changes in antioxidant enzyme activities, decreased activities of complexes (I, III, and IV) of the electron transport chain, and impaired mitochondrial respiration in PAD patients. Together, these data suggest the pathological implication of excessive oxidative stress in myofiber and PAD damage. Systemic inflammation is a pathophysiological mechanism of paramount importance in PAD and may contribute to skeletal muscle damage. Consequently, numerous vascular inflammatory markers, such as IL6, IL1 receptor antagonist, fibrinogen, and CRP, were elevated in PAD patients compared to controls [72]. Other research has shown a relationship between higher levels of these markers of the inflammatory process, a reduced ability in the 6-min walk distance, overall performance, and lower calf strength with more adverse characteristics. Therefore, it is possible to relate sarcopenia in these patients to the origin of the elevated inflammation [73,74].

The onset of the low-grade and chronic inflammatory process contributes to a failure in glucose, protein, and lipid metabolism, endothelial dysfunction, cardiovascular conditions, sarcopenia, and heart failure (HF). Outcomes for this condition are systemic, including exacerbated myokine release, the elevation of visceral fat, and pro-inflammatory status. Furthermore, tissue hypoxia, inflammation, and cellular apoptosis are associated with HF-related hemodynamic deficiencies. These events lead to impaired metabolism, a pro-inflammatory state, and chronic hypoperfusion that contributes to morpho-functional changes in different organs, leading to a general decrease in physiological reserves and an increase in vulnerability [75].

Based on the above, it is evident that not only frailty can trigger HF, but also HF can trigger frailty, which is associated with worsening physical condition, cognitive loss, and quality of life of these patients, arising from the overload of processes pro-inflammatory drugs and reduced tolerance to physiological stressors. The worsening of the patient’s general condition, such as the decrease in muscle mass, increase in adipocytes, and increase in lipids, are consequences of the exacerbated increase in inflammatory biomarkers and insulin resistance [76,77,78].

Type 2 diabetes mellitus (T2DM) and sarcopenic obesity (SO) are also closely related, as resistance to insulin leads to impaired uptake and use of cellular glucose, leading to numerous consequences such as obesity. It is also important to note that subjects with (T2DM) are more likely to have sarcopenia compared to euglycemic individuals [79]. Sarcopenia can contribute to the development and progression of T2DM due to the reduction of muscle mass and uptake of glucose, and the increase in localized inflammation, which can arise through inter and intramuscular adipose tissue accumulation. Furthermore, the increase in adipose tissue favors the production of several cytokines that increase muscle catabolism [64,80,81].

Not only aging but also a sedentary lifestyle contributes to obesity. Directly, as pointed out before, adipocytes generate the recruitment of macrophages leading to the release of leptin, chemerin, resistin, TNF-α, ILs, and interferon-γ (INF-γ), resulting in a low-grade inflammation scenario. Other studies have shown that the inflammatory process plays a significant role in the evolution of SO and in the morbidity and mortality induced by this condition. SO, in turn, has suppressive effects on muscle strength. In addition, with the increase in leptin, free fatty acids accumulate, stimulating the adipose tissue deposit in the liver, heart, and spleen. Decreased consumption of fatty acids is associated with increased generation of ROS. SO is also strongly related to the expression of various inflammatory transcription factors, such as nuclear factor-kB (NF-kB), which modulates proteolytic pathways and promotes inflammation. The low-grade inflammatory process and the ectopic distribution of fat cause myokine dysregulation and mitochondrial dysfunction. Myokines can elevate insulin resistance, and mitochondrial dysfunction causes an increase in lipid peroxidation, increasing inflammation, insulin resistance, and oxidative stress, further collaborating with the development of SO [82]. Considering T2DM and SO, it is necessary to prepare physicians, nutritionists, physiotherapists, and physical educators to recognize early sarcopenia and its risk factors in patients with T2DM and to perform appropriate therapeutic approaches capable of preventing and treating this condition [83,84,85,86,87,88].

Chronic obstructive pulmonary diseases (COPD), asthma, and pulmonary tuberculosis are among the pulmonary pathologies with systemic manifestations and can harm other systems, including skeletal muscle. These pathologies can cause systemic inflammation, decreased oxygenation, and oxidative stress, further degrading skeletal muscle with age [89]. Furthermore, the impairment of functional capacity and lack of physical exercise in patients with respiratory diseases also contribute to the appearance of sarcopenia in patients with advanced age. Exacerbated sarcopenia in these diseases can further accelerate muscle and respiratory degeneration and needs a more rigorous description for successful interventions. However, methods for assessing muscle decline are expensive, require an extended period, and may require radiation exposure, justifying the use of plasma biomarkers as a diagnostic method for sarcopenia [44].

Respiratory diseases increase the degradation of skeletal muscle with age. These diseases also cause an increase in systemic inflammation and OS, which can exclusively contribute to the reduction of skeletal muscle mass and strength, irrespective of the spirometry decline. A considerable link between muscle atrophy and weakness with markers of inflammation and OS in aging supports this. Therefore, searching for circulating biomarkers to predict muscle mass and strength under these conditions accurately remains challenging. However, studies have shown that skeletal muscle frailty was associated with greater chances of developing respiratory compromise. Conversely, a respiratory compromise was associated with greater chances of developing frailty [90].

Thus, there was a greater expression of markers of inflammation and OS in the three lung diseases. However, these markers are not specific and demonstrate general health rather than the health status of skeletal muscle and/or lungs. The glycoprotein Dickkopf-3 (Dkk-3) provides an advantage over these biomarkers through its expression in skeletal muscle and may be associated with systemic inflammation. Circulating levels of Dkk-3 are elevated with aging and in conditions involving cellular senescence, which has a role in cellular age-associated disorders in the basal epithelial cells of the prostate, suggesting that Dkk3 may have a role in aging-related disorders [44,91].

### 4.2. Nutritional Approach to Sarcopenia: The Role of Amino Acids, Omega 3, Vitamin D, and Calcium

Currently, the relationship between dietary patterns and the maintenance of skeletal striated musculature in the aging and health process is very discussed. The therapeutic and nutritional approach is a daily challenge and must be considered to provide patients with a superior quality of life and life expectancy. Nutrients such as protein, amino acids, vitamins, Omega 3, and calcium have been described as nutrients that can prevent sarcopenia when appropriately ingested [92]. Table 1 shows the clinical trials that investigated the use of supplements in sarcopenia.

#### 4.2.1. Amino Acids

##### Leucine

The approach taken on the effects of leucine supplementation in the sarcopenic elderly showed improvement in muscle protein synthesis [21]. Most studies show beneficial effects in increasing protein synthesis, maintaining or gaining lean mass, and increasing body weight. Some studies, however, confirmed these findings when supplementing with leucine, but when analyzing muscle strength, no gain was observed [20,93]. According to international guidelines, the recommended dosage of this supplementation is 3 g of leucine associated with 25–30 g of protein in the three main meals for the elderly population [93]. In addition, other studies also showed that this dosage was effective for treating these patients [21,94]. Other studies also proposed different dosages, even with a lower prevalence among the works. The PROT-AGE study group, according to their analysis of data in the literature, proposed, for elderly patients, a leucine supplementation containing about 2.5–2.8 g [93]. In elderly males without comorbidities (65–88 years), consumption of 5 g of leucine in each of the main meals for three days improved myofibrillar protein synthesis in the same way as for those with a protein intake of 0.8 g/kg of body weight/day, as in those with a dietary intake of 1.2 g protein/kg/day. This same study showed that 15 g of a high-protein drink with 4.2 g of leucine given twice a day for six days generated greater myofibrillar protein synthesis compared to the same drink with only 1.3 g of leucine [20]. However, although in most of the reviewed articles, the most prevalent dosage is 3.0 g per meal with a protein consumption of 25–30 g, the heterogeneity of the studies makes it impossible to reach definitive conclusions or even a consensus on the dosage administered. Still, it is possible to admit the beneficial effects of supplementation in sarcopenic patients or those at risk of developing the disease. Table 1 shows the clinical trials performed with this amino acid.

##### Branched-Chain Amino Acids (BCAA)

Much discussion has been raised about the effects of the branched-chain amino acids (BCAA: include valine, leucine, and isoleucine) supplementation as a therapy for sarcopenia, as it encompasses some pathophysiological mechanisms that contribute to the maintenance of lean mass in various pathologies, such as liver cirrhosis, for which, even though there is no consensus on which patients can benefit from supplementation or the ideal amount for each patient, its use is supported [22,95]. In a randomized trial, sarcopenic patients with cirrhosis of the liver were chosen to receive 12 g/day of oral BCAA or placebo for six months, plus 30 min/day of exercise, diet, and standard medical therapy. When analyzing the patients in the placebo group, it was concluded that adding BCAA associated with exercise did not significantly improve the functional measures of sarcopenia in patients with liver cirrhosis [22]. It is important to say that studies on the relationship between BCAA and aging-related syndrome, such as sarcopenia, generally present divergent conclusions about the benefits of BCAA supplementation.

BCAA may also benefit sarcopenic patients after a cerebral vascular accident. In one study, it was revealed that the group that was supplemented with BCAA, associated with intensive rehabilitation therapy, when compared to the control group, showed significant improvement in terms of skeletal muscle index and functional status, suggesting considering BCAA as adjuvant therapy [23]. In a trial performed in post-stroke patients, 3.5 g of amino acids (1.6 g of leucine, 0.9 g of isoleucine, 1.1 g of valine) and 6.5 g of protein were used, together with vitamin D (40 IU of per 125 mL), this supplementation was effective in decreasing the percentage of body fat and improving performance when combined with breakfast along with physical exercise [24].

To date, no solid conclusions about any therapeutic benefit of BCAA supplementation can be confirmed. After all, studies have used supplements rich in various amino acids and even other nutrients, so the effect of this supplementation cannot be proven as a benefit of BCAA. The mechanisms for any associations between BCAA levels, sarcopenia, and lean body mass are complex and depend on many factors. However, the use of BCAA as an integral part of protein intake is associated with improvements in muscle function in the elderly, as the effects of BCAA on protein synthesis and consequently on muscle mass depend on sufficient dietary protein and amino acid intake adequately and not the isolated use of BCAA [96].

Table 1 summarizes the clinical trials performed with tis compound.

##### Creatine

It is not entirely clear whether reductions in the body’s creatine occur as a direct consequence of aging or whether other modifiable factors, such as a decline in physical activity or nutritional intake, particularly protein, may also play a role. There is robust evidence showing the benefit of supplementing this compound on the ability to increase lean mass and muscle strength. However, it is less clear whether a direct effect of creatine causes these benefits or whether they are mediated by physical training. Data from most studies indicate that creatine alone is likely to result in little or no benefit for muscle strength, muscle mass, and functional performance. According to numerous studies, supplementing with this compound can increase lean mass and strength, reducing the risk of falls and bone mineral loss. In one study, it was concluded that the most efficient dosage to increase creatine accumulation in skeletal muscle would be 5 g administered 4 times/day, i.e., 20 g/day, for 5 to 7 days, and a maintenance dose of 3 to 5 g/day [97,98,99]. The consumption of other macronutrients (such as carbohydrates and proteins) in conjunction with creatine can bring about greater creatine absorption in the muscle. Furthermore, current studies found that this supplementation increased lean mass and muscle strength in conjunction with resistance training compared to exercise alone. [99]. Another study with a sample of men aged 65 on average who consumed a dose of several ingredients, including 2.5 g of creatine for 12 weeks associated with resistance training, showed a reduction in inflammatory processes compared to a placebo [100]. However, the effects of creatine supplementation on muscle mass without physical training were analyzed in several studies, and it was found that there was no positive effect on long-term use [98]. Although many studies show the benefit of creatine associated with physical exercise in maintaining lean mass, some studies diverge from this idea in older individuals. In addition, evidence of adverse events with creatine supplementation is not yet well-studied in the elderly [97]. For these reasons, clinical trials should be performed to clarify this lack in the information about the use of creatine in sarcopenic patients.

#### 4.2.2. Omega 3

Pro-inflammatory cytokines are directly associated with muscle wasting. Consequently, the anti-inflammatory effects of LCPUFAs (long-chain omega-3 polyunsaturated fatty acids omega-3) may be positive in preventing the waste of muscle mass and strength associated with aging, sarcopenia, and frailty [101]. In a study carried out with healthy older adults who were divided into two groups and followed for 8 weeks, group 1 received 4 g/day of corn oil, and group 2 received the same dose of ω-3 (eicosapentaenoic acid—EPA and docosahexaenoic acid—DHA). The results showed that in group 2, there was an increase in the rate of muscle protein synthesis (MPS). Another study using the same amount of DHA and EPA for six months carried out with elderly people of both sexes found that there was an increase in thigh muscle volume, grip strength, and muscle strength compared to a control group; however, regarding body weight, total body fat mass or intramuscular fat had no significant effect. In a study with supplementation of 1500 mg/d DHA and 1860 mg/d EPA over 6 months in elderly males and females, it was noted that there was an increase in thigh muscle volume, grip strength, and muscle strength in one repetition. It showed a trend towards increased mean isokinetic power compared to a control group. Still, body weight mass of total body fat or intramuscular fat had no significant effect and did not raise safety concerns. Regarding the impact of this fatty acid supplementation concerning the increase in lean mass, a study used the amount of 2 g/day, and it was found that after six months of intervention, there was an increase in lean mass and also an improvement in walking speed; on the other hand, those who received a lower dose did not show significant results. Other studies corroborate the association between increased protein synthesis, fighting catabolic events, and sarcopenia with ω-3 supplementation [101,102,103,104]. In contrast, a survey carried out with healthy elderly aged 65 to 84 years for 14 weeks, using a dose of 3 g/day of ω-3 together with resistance exercises, revealed muscle anabolism, catabolism, and inflammation were not affected by its consumption. There was no change in muscle activity or protein synthesis [25].

There is increasing evidence of a beneficial effect of omega-3 PUFA supplementation in sarcopenic older adults. However, the correct/exact dosage, frequency, and use (alone or combined) in the treatment and prevention of sarcopenia still need to be explored further [105].

Some authors suggest the intake of 1500 mg/day of DHA and 1860 mg/d EPA to obtain benefits such as increased protein synthesis, gait speed, muscle strength, and physical performance. The authors concluded that the lack of data from intervention studies hampers the development of recommendations on the intake of omega-3 LCPUFAs to prevent sarcopenia [26,101]. More randomized controlled trials, with different durations and doses, are needed to establish its effect on maintaining lean body mass in the elderly and decreasing the risk of sarcopenia

Table 1 shows the clinical trials performed with omega 3.

#### 4.2.3. Vitamin D

Vitamin D, also known by its active form 1,25-dihydroxycholecalciferol, or calcitriol, is an essential element for intestinal absorption of calcium, magnesium, and phosphate, in addition to being highlighted concerning the proliferation and differentiation of skeletal muscle cells and bone metabolism. Recent studies demonstrate that the deficiency or insufficiency of this vitamin is directly related to sarcopenia. Moreover, Vitamin D status can be a useful marker for predicting hip fractures, early death, total mortality, and the development of sarcopenia [106,107,108,109,110,111]. This vitamin has a role in the number and diameter of type II muscle cells, mainly type IIA. Type IIA cells induce fast muscle contraction, and are critical for short-duration, high-intensity exercise activities such as sprinting, jumping, acceleration, and deceleration [112].

According to the International Osteoporosis Foundation and other articles discussed, vitamin D intake of 800 to 1000 IU/day in the elderly benefits muscle performance, muscle strength, and decreased risk of fractures [111,113]. Although a single-dose standard for the treatment of sarcopenia has not yet been established, a recent study suggests the adoption of a dose of 800 IU/day [114].

On the other hand, single weekly or monthly doses of 300,000–600,000 IU have not shown improvement and are not recommended in the treatment of sarcopenia [115,116]. It is essential to consider that in individuals with normal levels of vitamin D, supplementation is usually not necessary since some data showed that the supplementation with this vitamin reduced general physical performance in a middle-old population (but not young-old or old-old populations) [114,117]. No clinical trials were included for the use of Vitamin D and sarcopenia, and for these reasons, it is not clear how this vitamin could be really effective for sarcopenia.

Figure 4 highlights the potential crosstalk between vitamin D, skeletal muscles, and bones in the development of sarcopenia.

#### 4.2.4. Calcium

Studies show that calcium is mainly responsible for muscle contraction strength [118]. Scott et al. [30] showed that higher calcium intake exerted positive increasing trends for increased appendicular muscle mass in community-dwelling older adults. Practically, the elderly usually present lower calcium intake, as well as lower calcium absorption and altered calcium homeostasis, which are all phenomena that link with muscle weakness in the aged muscles [119,120]. Physiologically, the skeletal muscles and the proper skeleton are superbly adapted to their functions: movement and equilibrium, combining strength with lightness, and calcium mineral are involved in skeletal muscle regulation and maintenance. Additionally, calcium contributes to myofibers’ neuromuscular command and regulation, such as the intracellular myosin proteins, which interact with actin and produce skeletal muscle contraction and relaxation. However, calcium does not work alone and depends on vitamin D to exert its action in skeletal muscles [121,122].

Besides the importance of calcium in promoting movement from muscles, as well as myosin and actin actions, the evidence regarding calcium supplementation alone reducing the decline of appendicular skeletal muscles mass and function is still weak, and therefore, there is a lack of evidence regarding whether sarcopenic individuals would be graced with the supplementation of this mineral. However, case reports and articles about proximal myopathies in lacto-vegetarian patients are easy to find, and supplementation with calcium and vitamin D is the resolving treatment [123]. Hirata et al. [124] were the first to report a case of elevated muscle enzymes in the serum of patients with severe hypocalcemia. Policepatil et al. [125] also reported a case in which hypocalcemia myopathy was derived from secondary hypoparathyroidism, and the resolving-treatment was also calcium supplementation.

In an animal model, Bennett et al. [126] evaluated the effects of calcium- β-Hydroxy- β-methyl butyrate supplementation (Ca-HMB) on muscle mass and function and aging-associated apoptotic signaling in unloaded but non-atrophied extensor *digitorum longus* muscles of aged rats. The animals were all gavaged daily with 170 mg of Ca-HMB. The results revealed little or no association between Ca-HMB supplementation and the rats’ muscle mass and function, as well as with neuromuscular signaling.

Burned patients are at elevated risk of developing hypovitaminosis D and osteopenia or sarcopenia. In a one-year pilot randomized controlled trial, Rousseau et al. [28] evaluated the roles of cholecalciferol supplementation and optimized calcium intake on muscle strength among adults with severe burns. The patients received a quarterly intramuscular injection of 200,000 IU of cholecalciferol and daily oral calcium. The results showed that the calcium combined with cholecalciferol effectively improved muscle health but not bone health, protecting the patients against sarcopenic stimuli.

Jabbour et al. [27] conducted a randomized, double-blind, controlled trial with older adults to evaluate the effects of high dosages of vitamin D supplementation on indices of sarcopenia and obesity among the studied patients. However, all the included subjects also received 1000 mg of calcium citrate daily. The results demonstrated that these interventions did not ameliorate the indices of sarcopenia or adiposity in older adults.

Stout et al. [29] conducted a randomized, double-blind, placebo-controlled pilot trial with men and women more than 65 years old to evaluate the effects of CaHMB (1.5 g daily) with and without resistance training on muscle health. The results showed that CaHMB significantly improved strength and muscle quality without resistance exercise. However, resistance exercise was considered superior in improving all body composition parameters and functionality.

As aforementioned, the hope of calcium supplementation in protecting against muscle atrophy and degradation is still obscure. The results of in vitro studies and human clinical trials are divergent, and some of the dosages used are not patronized internationally. In addition, calcium supplementation is not usually given individually but is associated with vitamin D, as they are functionally codependent. Therefore, studies regarding calcium supplementation in protecting against sarcopenic myopathies are needed as it could be a promising therapeutic option against sarcopenia.

### 4.3. Medication Approach to Sarcopenia

Although lifestyle interventions (physically and nutritionally) have shown a profound impact on sarcopenia treatment, several different drugs have been tested with various scientific evidence levels to impact skeletal muscle maintenance among diseased patients. These can alter the sarcopenic’s metabolic parameters, protect against cardiovascular diseases and outcomes while protecting muscles, or even act directly on them [127,128]. Some of these drugs are sodium–glucose cotransporter 2 (SGLT2) inhibitors, metformin, growth hormone, glucagon-like peptide-1 receptor agonists (GLP-1A), statins, losartan, and peptidyl peptidase 4 (DPP-4) inhibitors.

#### 4.3.1. Sodium–Glucose Cotransporter 2 (SGLT2) Inhibitors

SGLT2 inhibitors are medicaments with well-established nephroprotective and cardiovascular protection functions. These antidiabetic drugs are highly associated with improved glycemic controls and slimming and anti-hypertensive actions. However, few studies have emerged highlighting the possible roles of SGLT2 inhibitors against muscle atrophy and in improving skeletal muscles’ function [129,130].

In an in vivo study with db/db mice, Bamba et al. [131] evaluated the possible role of luseogliflozin, an SGLT2 inhibitor medication, in preventing skeletal muscle atrophy. Eight-week-old male mice were used and fed with a standard diet added or not to luseogliflozin at 0.01% *w/w* in chow for approximately eight weeks. The results highlighted that the lipidome of the rats was modified by the luseogliflozin’s treatment with decreased intramuscular fatty acid metabolism markers (such as Scd1, Fasn, and Elovl6) and genetic muscle atrophy-related transcripts (such as Mstn, Trim63, Fbxo32, and Foxo1) expression that has not only decreased the mice’s visceral fat accumulation but also increased the soleus muscles weight of the treated mice.

Otsuka et al. [132] differentiated the effects of canagliflozin, another SGLT2 inhibitor medication, on slow and fast skeletal muscles from mice that are nondiabetic models. In this experiment, the rats were treated with canagliflozin, and the results showed that the fast muscles’ functions increased with the rats’ increased food intake, whereas slow muscles demonstrated unaffected functions. Both muscle contents were maintained, and biochemical analyses revealed that glycolytic metabolites and adenosine 5′-triphosphate (ATP) were increased in fast muscles. In the slow muscles, ATP was maintained, although glycolytic metabolites were reduced following the canagliflozin’s treatment. Amino acids and free fatty acids are nutrients that were increased in slow muscles but unchanged in fast muscles. All these results promise to bring new insights to treating patients at risk of sarcopenia.

Among human patients, Sasaki et al. [31] demonstrated that luseogliflozin in a 52-week treatment with moderately obese Japanese type 2 diabetes mellitus (T2DM) patients was associated with favorable metabolic and body composition changes. In this study, minimal skeletal muscle loss was observed.

#### 4.3.2. Growth Hormone (GH)

GH is the basis of human physiological growth. Although in recent years, the recombinant human GH (rHGH) has become a target of abuse in the community of competitive sports, the variations in human and animal growth have awakened scientists’ attention. As it is also related to muscle development, GH emerged as a potential therapeutic against sarcopenia [133,134]. The GH depends on its carrier insulin-like growth factor (IGF-1) to act physiologically and therapeutically. Bian et al. [135], in a cohort of 3276 elderly patients, evaluated that both GH and IGF-1 serum concentrations were associated with sarcopenia. In this study, the elderly’s appendicular skeletal muscle mass was positively correlated with GH and IGF-1, but negatively with high-density lipoprotein cholesterol (HDL-c).

Other studies have indicated that GH deficiency may lead to increased myostatin levels, which is an atrophic factor, with dissociation in autocrine IGF-1 effects on muscles’ protein synthesis. These are pro-sarcopenic phenomena [136]. In adults with human immunodeficiency virus (HIV), Adrian et al. [32] evaluated the role of tesamorelin, a GH-releasing hormone analog, in muscle mass. The results showed that among those treated patients with a clinically significant decrease in the visceral adipose tissue, tesamorelin effectively increased the density and area of the skeletal muscles, which are all anti-sarcopenic actions. Table 2 summarizes the studies performed with medications in sarcopenic patients.

#### 4.3.3. Glucagon-Like Peptide-1 Receptor Agonists (GLP-1A)

Glucagon-like peptide-1 (GLP-1) corresponds to a human incretin hormone that was in the past frequently associated with only the treatment of T2DM and its related conditions, such as obesity. This molecule is derived from the proglucagon molecule, and nowadays, it is known to affect the pancreas, liver, brain, heart, gastrointestinal organs, adipocytes, and last, but not least, the skeletal muscles [137].

Hong et al. [138] conducted an in vitro study with C2C12 mouse-derived myotubes to evaluate whether exendin and dulaglutide, which is a GLP-1A, in the presence or absence of dexamethasone could be associated with the regulation of muscle atrophic factors’ expression. The researchers found that the injection of 100 ng/day intraperitoneally of exendin for 12 days and 1 mg/kg/week subcutaneously of dulaglutide for 3 weeks ameliorated muscle wasting and enhanced myogenic factors through the suppression of myostatin and the up-regulation of the GLP-1A-receptor signaling pathways, respectively. Exendin also down-regulated the expression of other muscle atrophic factors, such as the F-box only protein 32 (atrogin-1) and muscle RING-finger protein-1 (MuRF-1) in the dexamethasone-treated mice.

In another study, Khin et al. [139] investigated whether dulaglutide could present a therapeutic potential against muscle wasting due to aging in mice. The researchers affirmed that the GLP-1A improved muscle mass and muscle strength in the aged mice, helping improve histopathological parameters by an increased shift toward middle and large-sized skeletal muscle fibers. These results can be associated principally with the dulaglutide-derived activation of the mitochondrial biogenesis regulator peroxisome proliferator-activated receptor-gamma coactivator-1α (PGC-1α). Dulaglutide’s treatment decreased the expression of the atrophic muscle factors myostatin, muscle RING-finger protein-1, and atroogin-1, and increased the expression of the myogenic factor MyoD, also exerting anti-inflammatory actions in the aged mice by decreasing the serum levels of the pro-inflammatory cytokines interleukin 6 (IL-6) and tumor necrosis factor-alpha (TNF-α).

Among overweight and obese T2DM patients, Perna et al. [33] investigated the roles of liraglutide, another GLP-1A, against fat accumulation and muscle loss to prevent sarcopenia. Data from nine patients aged between 68.22 ± 3.86 years were evaluated, and the results showed that 24 weeks of liraglutide treatment could effectively cause a reduction in the fat mass while preserving skeletal muscles’ tropism, therefore protecting against sarcopenia.

#### 4.3.4. Metformin

Metformin is an effective blood-glucose-lowering agent that acts directly against insulin resistance. Although the liver has been elected as the major site of metformin’s effects, many other organs can respond to this drug, especially the gut. Recently, researchers highlighted the anti-cancer and anti-aging potential of metformin; therefore, much has been said about its potential against sarcopenia, but with some controversial results [140,141].

Kang et al. [142] demonstrated in an in vitro study that metformin’s treatment could impair muscle function by up-regulating myostatin in skeletal muscle cells through the activation of the histone deacetylase 6 (HDAC6) and forkhead box O3a (FoxO3a) signaling pathways, the respective adenosine monophosphate-activated protein kinase-FoxO3a-HDAC6 (AMPK-FoxO3a-HDAC6) axis. These results had stronger evidence among metformin-treated wild-type rats than in db/db mice.

Chen et al. [34], in a population-based study that comprised 1732 elderly patients with T2DM, demonstrated that metformin acted as a protective agent against the development of sarcopenia in this cohort of patients. However, these protective effects of metformin were also revealed in the elderly who combined drugs for different health problems, so the results could have been denigrated due to this fact.

Toledo-Pérez et al. [143], in an animal study with middle-aged Wistar female rats, evaluated the roles of metformin and tert-butylhydroquinone treatments combined with exercise in preventing the osteosarcopenic obesity. The results showed that combined metformin-tert-butylhydroquinone-exercise treatment could effectively increase the muscle mass and muscle strength of the treated mice, as well as decrease their body weight and fat percentage, improving their redox statuses and augmenting their survival.

In a glutaredoxin-1 (Grx1) KO mice study, Yang et al. [144] evaluated the role of metformin in ameliorating rats’ skeletal muscle atrophy. The results showed that metformin was efficient in regulating the rats’ intramuscular lipid accumulation and glucose utilization, therefore protecting them against aging-related conditions, especially sarcopenia, due to the Grx1 pharmacological approach.

Among ovariectomized mice, Zakeri et al. [145] studied whether metformin’s treatment could be associated with increased pro-cognitive and anti-sarcopenic benefits over one year of 1 and 10 mg/kg, daily treatment. The results showed that metformin increased the rats’ physical strength and longevity, probably due to the up-regulation of the brain-derived neurotrophic factor (BDNF).

Dungan et al. [146] studied the skeletal muscle of aged mice to evaluate whether metformin’s treatment could be effective in normalizing the growth signals of these muscles. The results showed that it was ineffective at normalizing those signals, therefore, not demonstrating effects pro-sarcopenia due to the hyperactivation of the mammalian target of rapamycin complex 1 (mTORC1).

In a double-blind, randomized, controlled clinical trial among non-diabetic pre-frail elderly patients, Laksmi et al. [35] evaluated the effects of metformin on handgrip strength, myostatin serum levels, and health-related quality of life. The results showed that 3 × 500 mg metformin for 16 weeks could effectively and significantly improve quality of life but did not improve handgrip strength or myostatin serum concentrations of the treated individuals.

In another in vivo study with old female Wistar rats, Hernández-Álvarez et al. [147] evaluated whether long-term metformin combined with moderate exercise could induce anti-sarcopenic effects. The results showed that this treatment induced a hermetic response that prevented the rats from losing strength and muscle mass. These accomplishments were associated with mechanistic redox state modulations by the metformin plus exercise routine.

#### 4.3.5. 3-Hydroxy-3 Methylglutaryl Coenzyme A Inhibitors (Statins)

Statins are used worldwide for cardiovascular disease prevention. These medications decrease blood levels of lipids and the development of atherosclerosis. However, given the usual long-term use of statins for cardiovascular protection by patients at risk, much has been studied about how statins can affect the body’s homeostasis and lead to diseases other than cardiovascular [148,149].

Statins can present roles in muscle loss. These medications are considered pro-sarcopenic due to various derived actions corresponding to connected IGF-1 deficiency, inflammation, and the activation of the ubiquitin-proteasome system as the major non-lysosomal intracellular protein degradation system. Studies have also postulated that statins, through their involvement in the ubiquinol-proteasome pathway, can provoke myocytes’ membrane stability, which leads to the consequent changes in protein degradation machinery due to increased intracellular proteolytic cascades [150,151]. However, not all types of sarcopenia appear to have the same involvement of statins.

In a population-based study, Lin et al. [37] evaluated the role of statin for the risk of newly diagnosed sarcopenia among chronic kidney-diseased patients. The results demonstrated that patients with chronic kidney disease were not affected muscularly by the statin treatment received. Controversially, these patients could receive statins at higher dosages to reduce the incidence of newly diagnosed sarcopenia. These results indicate that much more clinical trials must be developed to science truly evaluates statins’ use for sarcopenia occurrence or prevention.

Lindström et al. [36] also found controversial results about statin use on sarcopenic-related outcomes. In a cohort of patients that underwent endovascular aortic repair, these authors evaluated whether statin use could be associated with the development of sarcopenia and long-term survival in those patients. The results showed that using statins was associated with increased survival and reduced long-term mortality without predisposing to increased sarcopenia.

In a cohort of 639 dwelling older men and women, Witham et al. [38] demonstrated that the use of angiotensin-converting enzyme inhibitors, thiazides, or statins was not associated significantly with differences in grip strength decline.

#### 4.3.6. Losartan

Losartan acts as an angiotensin II receptor antagonist that possesses potent anti-hypertensive and also cardioprotective effects during heart failure. This anti-hypertensive is widely commercialized across the planet, and its production is even larger each year [149,150]. The uses and applications of this drug have been extensively augmented during the last years, and this medication was found to be useful against sarcopenia. This anti-hypertensive is widely commercialized across the planet, and its production grows even larger each year [152,153].

Burks et al. [154] found out in an in vivo study with sarcopenic mice that losartan could effectively restore the mice’s skeletal muscle remodeling and protect against atrophy by disuse during sarcopenia development. It is well-established that the increased transforming growth factor-β (TGF-β) signaling can contribute to the aged muscles’ satellite cells impairment, which leads to muscle repair difficulties. In turn, losartan exerts antagonism against the TGF-β signaling; therefore, this medication could be useful in protecting against muscle loss. In this study, losartan attenuated muscle loss inhibiting the canonical TGF-β signaling cascade and modulating the non-canonical. Additionally, immobilized mice treated with losartan were protected against muscle loss effectively by losartan’s mediated IGF-1/protein kinase b/mTOR (IGF-1/Akt/mTOR) signaling pathway activation. Therefore, it can be addressed that the use of losartan significantly improved muscle remodeling and protected against tissue disuse.

Lin et al. [155] demonstrated the effects of losartan and exercise on muscle endurance and mass in a model of old mice. These authors suggested that through principally antioxidant effects, losartan could effectively protect the mice from muscle loss and promote adaptations for exercise training endurance. In turn, Pahor et al. [156] demonstrated in a human study that losartan plus fish oil treatment prevented mobility loss in older adults due to anti-inflammatory effects on reducing IL-6 levels. Lastly, Takeda et al. [157] in cirrhotic rats demonstrated that losartan added to branched-chain amino acids treatment could effectively decrease skeletal muscle atrophy.

#### 4.3.7. Dipeptidyl Peptidase 4 (DPP-4) Inhibitors

DPP-4 inhibitors have been first associated with the treatment of T2DM. These medications can indirectly stimulate insulin secretion and inhibit glucagon production by elevating the levels of GLP-1 in the bloodstream. DPP-4 inhibitors have already replaced sulfonylureas as the second option for T2DM treatment and have also been investigated against other conditions, such as sarcopenia [158,159].

In patients with T2DM, Bouchi et al. [40] evaluated the roles of DPP-4 inhibitors on skeletal muscle mass loss. In a cohort of 37 patients treated with DPP-4 inhibitors, the results showed that these medications could effectively and significantly prevent progressive muscle mass loss with the aging of the treated patients.

In a 6-month follow-up study with older diabetic patients, Sencan et al. [39] found out that adding a DPP-4 inhibitor to the patients’ treatments could effectively and significantly result in a positive effect on muscle strength, counteracting T2DM-related sarcopenia occurrence.

Lastly, Rizzo et al. [41] comprised data from 80 elderly diabetic patients and found out that the use of DPP-4 inhibitors resolutely consisted of the idea that this class of medications can be used for the prevention of muscle mass loss among diabetic patients.

## 5. Limitations of the Included Studies

The studies included in this review have several limitations, such as those presented in Table 3 and Table 4. These limitations refer mainly to the absence of a placebo group in the investigations, inappropriate randomization, lack of sample calculation, and insufficient intervention time.

## 6. Conclusions

Sarcopenia is a multifactorial disorder related to the loss of muscle mass due to aging and can worsen or be aggravated by other pathologies such as obesity, diabetes, and cardiovascular diseases. It is widely accepted that physical training should be a fundamental component of sarcopenia treatment and management strategies. However, the role of nutritional supplementation, especially of isolated nutrients, is controversial. Some nutritional support strategies may be indicated as prevention or recovery from sarcopenia. There is sufficient evidence for some specific dietary components, for example, adequate daily protein intake and the distribution of protein consumption throughout the day, that can maximize the anabolic response. Although with controversial results, medications such as metformin, GLP-1, losartan, statin, growth hormone, and dipeptidyl peptidase 4 inhibitors can also have been considered and can alter the sarcopenic’s metabolic parameters, protect against cardiovascular diseases and outcomes while protecting muscles.

We believe that in the near future, studies will shed light on when and if nutritional therapy should be used. Although the combined use of drugs and nutrients is beneficial, we emphasize that if this use is combined with physical activity (when possible), superior beneficial effects, including systemic effects, will be observed. In this sense, it is possible that, in some cases, the patient can benefit from adequate nutritional supplementation even before the disease is diagnosed, thus acting in a preventive and personalized way. However, there is a need to develop new clinical trials for evaluating nutrition and pharmacological therapy effects with a particular focus on patients with different types and stages of sarcopenia to assess interventions’ roles and outcomes.

## Figures and Tables

**Figure 1 biomedicines-11-00136-f001:**
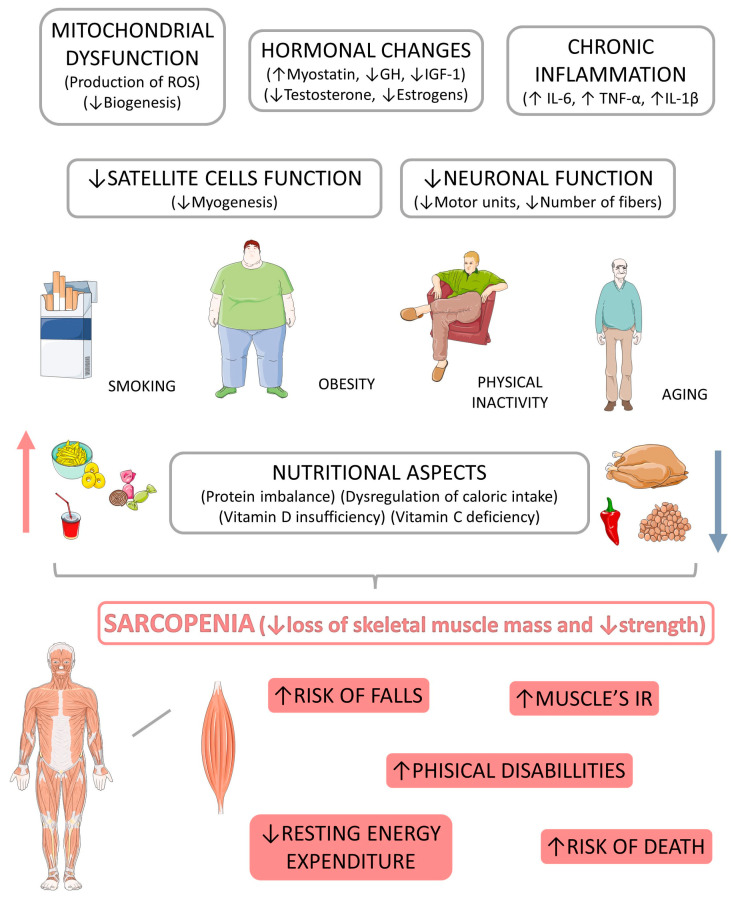
Sarcopenia-related risk factors and sarcopenia-related outcomes. ↑, increase; ↓, decrease; GH, growth hormone; IGF-1, insulin-like growth factor-1; IL-1β, interleukin 1 beta; IL-6, interleukin 6; IL, insulin resistance; ROS, reactive oxygen species; TNF-α, tumor necrosis factor-alpha.

**Figure 2 biomedicines-11-00136-f002:**
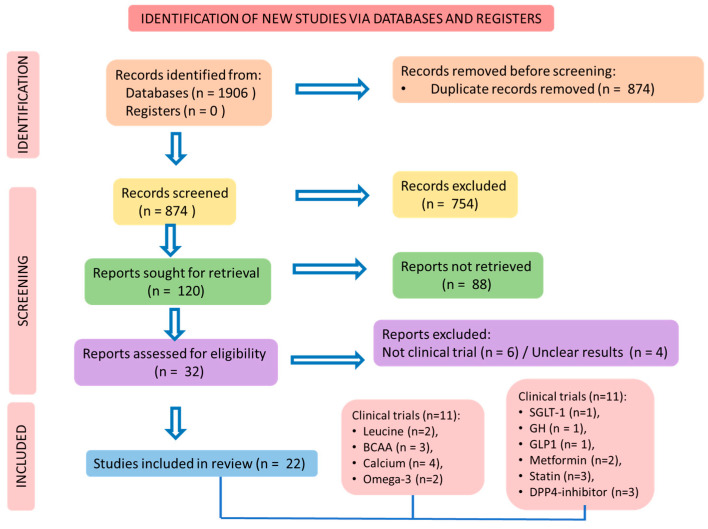
Flowchart showing the study selection of the studies.

**Figure 3 biomedicines-11-00136-f003:**
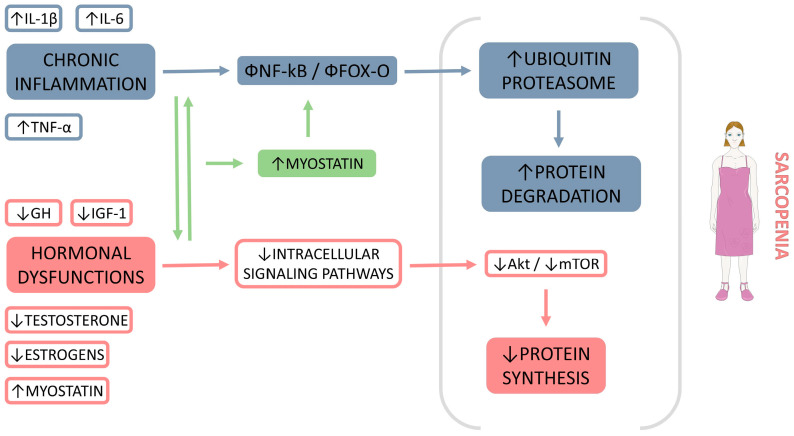
Relationships between chronic inflammation, hormonal dysfunctions, and sarcopenia. ↑, increase; ↓, decrease; Akt, protein kinase b; FOX-O, forkhead box O proteins; GH, growth hormone; mTOR, mammalian target of rapamycin; IGF-1, insulin-like growth factor-1; IL-1β, interleukin 1 beta; IL-6, interleukin 6; NF-kB, nuclear factor kappa b; TNF-α, tumor necrosis factor-alpha.

**Figure 4 biomedicines-11-00136-f004:**
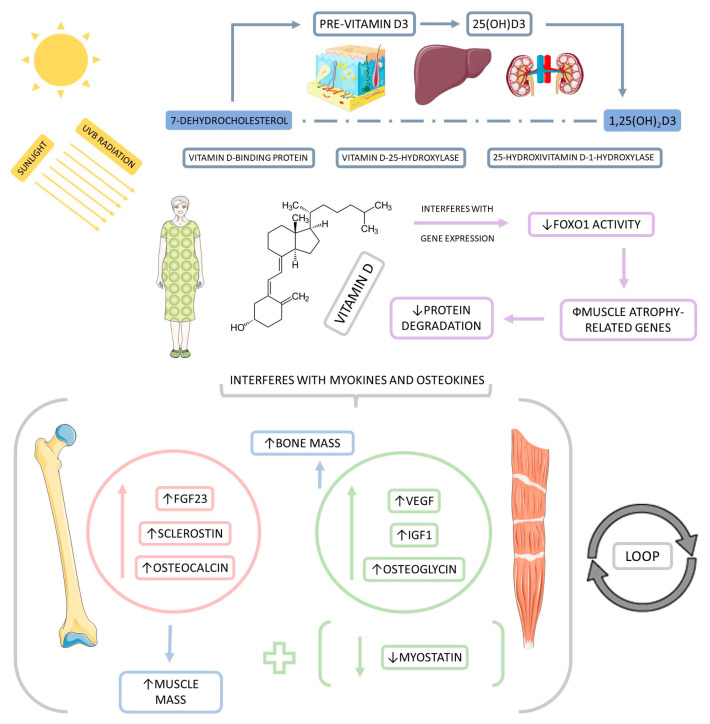
Vitamin D, skeletal muscles, and bones perform crosstalk in sarcopenia development. ↑, increase; ↓, decrease; FGF23, fibroblast growth factor 23; FOXO1, forkhead box protein 1; IGF-1, insulin-like growth factor-1, UVB, ultraviolet B; VEGF, vascular endothelial growth factor.

## Data Availability

Not applicable.

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
