# Peer review of "Insights into Pathogenesis, Nutritional and Drug Approach in Sarcopenia: A Systematic Review"

_biomedicines, 2023, doi:10.3390/biomedicines11010136_

Round 1

Reviewer 1 Report

I would like to acknowledge the effort on conducting this review. However, under my opinion there are various important concerns that make me decline towards rejection:

- The concept of sarcopenia is mainly based on loss of muscle mass (this is mentioned all over the mansucript). Nowadays, sarcopenia includes both the loss of strenght and muscle mass. 

- methods section is scarce and with low quality. 

- results of the review have not been shown.

- Therefore the discussion section is not properly understood. 

Overall it lacks scientific method and quality. Maybe following to some extent the  Preferred Reporting Items for Systematic Reviews and Meta-Analyses (PRISMA) cheklist and the PICO format to conduct the research could help.

Author Response

Reviewer 1

I would like to acknowledge the effort on conducting this review. However, under my opinion there are various important concerns that make me decline towards rejection:

Response: Dear Doctor, thank you very much for reviewing our manuscript. We would like to ask you to see all the modifications we performed to improve the quality. Besides that, we asked a native English speaker to correct the manuscript. For these reasons, we humbly ask that you reconsider accepting the article.

- The concept of sarcopenia is mainly based on loss of muscle mass (this is mentioned all over the mansucript). Nowadays, sarcopenia includes both the loss of strenght and muscle mass. 

Response: Dear Doctor, thank you for this comment. We agree with you and included muscle strength in the definition of sarcopenia. Please see in the Introduction section, page 1, lines 21, 38-41, and also in the Discussion of pathophysiological aspects (lines 153-155 and 189-191).

- methods section is scarce and with low quality.

Response: Dear Doctor, thank you very much for this comment. We improved the description of this section. Please see page 5, lines 123-137 and pages 6-11.

- results of the review have not been shown. Therefore the discussion section is not properly understood. 

Response: Dear Doctor, thank you very much for this observation. We improved the description of this section. Please see page 5, lines 123-133, and pages 6-11. We build two tables with the included studies. Table 1 summarizes the studies with the use of supplements and Table 2 shows the trials that investigated the use of medications in sarcopenia. Please, see on page 6-11.

Overall it lacks scientific method and quality. Maybe following to some extent the  Preferred Reporting Items for Systematic Reviews and Meta-Analyses (PRISMA) cheklist and the PICO format to conduct the research could help.

Response: Dear Doctor, thank you so much for this comment. We included the PRISMA guidelines to improve the search and inclusion of the studies. This inclusion was based on PICO.  Please see pages 3-4, lines 83-119. Moreover, we included PRISMA flowchart to show the selection of the studies and the inclusion of the trials.

            Dear Doctor, we know that your time is precious. Thank you very much for correcting our manuscript.

Reviewer 2 Report

The manuscript entitled " Sarcopenia: Insights Into Pathogenesis, Nutritional and Drug Approach" is focused in a revision of the nutritional and drug approaches used in an attempt to improve the health and quality of life of sarcopenic patients. The topics described in this revision is of utmost importance, and with just a few improvements, it is suitable to be published.

Considerations:

- Authors performed a search in databases that resulted in the selection of several studies. Therefore, the selected studies, or at least the most important studies, must be showed in the results section. Therefore, authors should present a table with the selected studies and the most important information collected from them.

 - The section “Sarcopenia: pathophysiological aspects” describes more than the pathophysiological aspects, in the sense that it correlates sarcopenia with other disease. Therefore, herein a new section should be inserted, in which the correlation between sarcopenia and other diseases would be described.

 - In the discussion section, the authors should further hypothesize the mechanisms associated with each intervention described. They only do so for some of them.

Author Response

The manuscript entitled " Sarcopenia: Insights Into Pathogenesis, Nutritional and Drug Approach" is focused in a revision of the nutritional and drug approaches used in an attempt to improve the health and quality of life of sarcopenic patients. The topics described in this revision is of utmost importance, and with just a few improvements, it is suitable to be published.

Considerations:

- Authors performed a search in databases that resulted in the selection of several studies. Therefore, the selected studies, or at least the most important studies, must be showed in the results section. Therefore, authors should present a table with the selected studies and the most important information collected from them.

Response: Dear Doctor, thank you very much for this observation. We build two tables with the included studies. Table 1 summarizes the studies with the use of supplements and Table 2 shows the trials that investigated the use of medications in sarcopenia. Please, see page 6-11.

 - The section “Sarcopenia: pathophysiological aspects” describes more than the pathophysiological aspects, in the sense that it correlates sarcopenia with other disease. Therefore, herein a new section should be inserted, in which the correlation between sarcopenia and other diseases would be described.

Response: Dear Doctor, thank you very much for this comment. We included a new section for diseases related to sarcopenia. Please, see line 245.

 - In the discussion section, the authors should further hypothesize the mechanisms associated with each intervention described. They only do so for some of them.

Response: Dear Doctor, thank you very much for this observation. Please, see the modifications we performed in the manuscript, for example, in lines 432-434, 470-472, 496-498, and 559-561.

Dear Doctor, we know that your time is precious. Thank you very much for reviewing this manuscript.

Round 2

Reviewer 1 Report

I congratulate the authors for the improvements on the manuscript. In relation the methods section, if you have used the PICO format to define the research question, it should be described, which population was selected?, which intereventions? which comparisons? and which outcomes?

Author Response

   Dear Doctor,

   We very much appreciate your suggestions and your time in reviewing this manuscript.
   We re-adapted the tables to comply with the PICO format. Please see pages 6-11. We hope that these modifications are in line with your correction.

   Thanks so much again for your time.
    We wish you a wonderful 2023.